# A Comparative Analysis of the Grafting Efficiency of Watermelon with a Grafting Machine

Huan Liang [1], Juhong Zhu [1], Mihong Ge [1], Dehuan Wang [1], Ke Liu [1], Mobing Zhou [1], Yuhong Sun [1], Qian Zhang [2], Kai Jiang [3],* and Xianfeng Shi [1],*

[1] Wuhan Academy of Agricultural Sciences, Wuhan 430070, China; lianghuanconf@126.com (H.L.); hongye408@163.com (J.Z.); gmh917@126.com (M.G.); wdhuan1987@163.com (D.W.); cokelk@163.com (K.L.); zmb19682022@163.com (M.Z.); sunyh68@163.com (Y.S.)

[2] Research Center of Information Technology, Beijing Academy of Agriculture and Forestry Sciences, Beijing 100097, China; zhangq@nercita.org.cn

[3] Research Center of Intelligent Equipment, Beijing Academy of Agriculture and Forestry Sciences, Beijing 100097, China

[*] Correspondence: jiangk@nercita.org.cn (K.J.); shixf124@163.com (X.S.); Tel.: +86-10-5150-3504 (K.J.); +86-10-6538-6852 (X.S.)

**Abstract:** The rising age of the population in rural China and the labor intensity of grafting have resulted in a decrease in the number of grafters and a subsequent increase in their wages. Manual grafting can no longer satisfy the increasing demand for watermelon-grafted transplanting; thus, machine grafting will be an effective alternative. In order to accelerate the implementation of machine grafting in China, a comparative analysis between the automatic grafting machine (model 2TJGQ-800) and traditional hand grafting was conducted. The reliability and feasibility of machine grafting were evaluated through a comprehensive evaluation of the production capacity and grafting seedling quality. This study focuses on the grafting application of watermelon plug-tray seedlings. The scion and rootstock seeds were sown on 9 November 2022. Grafting experiments using an automatic grafting machine, skilled workers, and ordinary workers were conducted with the root-pruned one-cotyledon grafting method on 24 November 2022. The results showed that the machine grafting had a high uniformity and grafting speed. The grafting speed of the grafting machine was 774 plant·h$^{-1}$ and 1.65–2.55-fold higher than the hand grafting. With training, workers can improve their grafting speed, but it will still be slower than machine grafting. In addition, there was no significant difference in the grafting survival rate between the machine grafting and hand grafting. However, using machine grafting, the success rate decreased from 100% to 90.07% and the rootstock regrowth rate increased from 18.44% to 72.69%. Incomplete rootstock cutting, clip supply failure, and grafting drop failure are the three main factors that result in machine grafting failure. In conclusion, the grafting machine has advantages in terms of grafting speed and uniformity. Upon improving the accuracy of the cutting mechanism and grafting success rate, it will be adopted by commercial nurseries.

**Keywords:** watermelon; grafting efficiency; grafting machine; one-cotyledon grafting; comparison

## 1. Introduction

Grafting is an asexual propagation method that allows for two plant segments to be joined together, only for them to later grow and develop as a single composite plant. Grafting is widely implemented in cucurbit vegetable production to control soil-borne diseases caused by bacteria, fungi, oomycetes, viruses, or root-knot nematodes in Japan, China, Korea, and some parts of European and Asian countries [1,2]. Moreover, it can improve nutrient transport and resistance to abiotic stresses [3–5]. Grafting can also promote the growth and development of the grafted plants, increase the plant yield, and positively influence the fruit quality [6,7].

Despite the above advantages, their high cost is one main factor limiting the utilization of grafted seedlings on a large scale [8]. In nurseries, the cost of hand-grafted plants is estimated at approximately CNY 0.58 ¥ in China, compared to CNY 0.32 ¥ for non-grafted plants [9]. However, the costs of hand-grafted plants in Japan, Korea, and the USA were higher than those in China, where they can accumulate three to four times extra costs without grafting [10–12]. The cost of hand-grafted plants includes the cost of grafting operations, rootstock seeds costs, clip costs, and energy costs at grafted seedling healing, etc. It is estimated that the grafting process itself can amount to approximately one quarter of the total costs per grafted plant [13].

At present, grafting is mainly reliant on Chinese laborers. The hole-insertion method is the most widely used method for cucurbit grafting [14,15]. Its grafting speed is approximately 350 plants·h$^{-1}$ per worker. This method requires a highly skilled grafter and similarly high-quality scion and rootstock seedlings. At the time of grafting, the diameter of the scion must be smaller than the diameter of the rootstock stem, so that the scion can be inserted into a hole made between the two cotyledons of the rootstock. A smaller scion stem is easier to soften after cutting. It is difficult to insert the slender and soft stem into the hole of the rootstock. Moreover, rootstock regrowth commonly occurs after grafting using the above grafting method, which can result in graft failure or a decrease in yield, as the rootstock competes with the scion for water and nutrients [16]. To solve this problem, labor is required to remove the rootstock regrowth and damage to seedlings can occur [17,18]. Furthermore, the aging agricultural population and intensive workloads of grafting have led to a decrease in the number of available grafters. However, grafting is a task that requires considerable time. With the increasing demand for grafted seedlings, hand-grafting—which requires a large number of laborers—can no longer meet this demand.

Grafting machines, which can reduce the demands on hand-grafting laborers, expand production capacities, and improve product uniformity, have been proposed worldwide for many years. Since the 1980s, when researchers in Japan and Korea first began to conduct melon-grafting machine technology research, many types of grafting machines have been launched [19–21]. In 2010, ISEKI & CO., LTD. launched the GR803-U grafting robot in Japan (700 Umaki-cho, Matsuyama-shi, Ehime-ken, 799-2692 Japan). The robot was designed with a seedling linear cutting mechanism. It needed two workers to complete the seedling supply. Its productivity reached 800 plants·h$^{-1}$ and its survival rate reached 95% [22]. Helper Robotech Co., Ltd. launched the AFGR-800CS grafting robot in Korea (93, Hitech-ro, Jinrye-myeon, Gimhae-si, Gyeongsangnam-do, Korea). This grafting robot used the rotary cutting mechanism. It also needed two workers to complete the seedling supply. Its productivity reached 800 plants·h$^{-1}$ and its survival rate reached 95% [23,24]. Atlantic Man. SRL developed the Gr300/3 grafting robot in Italy (42024 Castelnovo di Sotto (RE), Italy). A single worker could complete the seedling supply. Its production efficiency was 300 plants·h$^{-1}$ and the survival rate of the grafted seedlings was 98% [25]. However, the high cost of these robots, the strict requirements for the agronomic traits of the scion and rootstock seedlings, and the complexity of their repairs are obstacles to their adoption in China [26]. China began researching grafting machines in the 1990s [27–29]. In 2005, China Agricultural University launched the 2JSZ-600 grafting robot, which used a double-arm bidirectional cutting mechanism. It adopted the operation mode of double-station feeding for the seedlings. Its productivity reached 600 plants·h$^{-1}$ and its survival rate reached 95% [27]. In 2011, the first commercial grafting machine (TJ-800, the National Agricultural Intelligent Equipment Engineering Technology Research Center, Beijing, China) was launched [30]. This robot was designed with a rotary cutting machine, which could precisely adjust the cutting angle. Two workers could complete the seedling supply, the productivity reached 800 plants·h$^{-1}$, and the success rate reached 95% in the laboratory. In order to improve the machine grafting accuracy, the improved and upgraded model 2TJGQ-800 was developed [31,32]. The grafting matching model was designed, which could adjust the grafting angle of the rootstock or scion. The matching rate of the rootstock incision

and the scion incision reached 98.03% [33,34]. The properties of this grafting machine are being continuously improved, but their current use for cucurbit is relatively low [35]. In China, the demand for melon grafting seedlings is approximately 50 billion, and millions of grafters working to produce grafting seedlings could meet this demand. The demand for machine grafting becomes more and more urgent with the reduction in grafters. However, the related research and development institutions have not given a reasonable and accurate answer to whether a grafting machine could replace manual grafting at the present stage. The lack of a feasibility analysis of machine grafting technology means that farmers and seedling nurseries cannot adopt machine grafting.

To ensure that machine grafting can be implemented in China as soon as possible, this study was conducted to examine the grafting efficiency and grafting seedling quality of a grafting machine compared with those of skilled grafters and unskilled grafters, based on the grafting machine 2TJGQ-800 (the National Agricultural Intelligent Equipment Engineering Technology Research Center, Beijing, China). This study focuses on the subject of watermelon plug-tray seedlings. The grafting was conducted using the root-pruned one-cotyledon grafting method. These results will highlight the existing problems of grafting machines and provide references for grafting machine improvement and optimization, contributing to the complete mechanization of grafting in China.

## 2. Materials and Methods

### 2.1. Seedling Production

The experiment was conducted at the Wuhan Agricultural Academy, Central China (30°27′ N, 114°20′ E, and altitude 22 m above sea level), between November and December 2022. The watermelon 'Zaojia84-24′ (*Citrullus lanatus*, Xinjiang Seed Co., Ltd., Urumqi City, China) was used as a scion and the interspecific pumpkin 'Zhenzhuang' (Jingyan Yinong, Seed Sci-Tech Co., Ltd., Beijing, China) was used as the rootstock. The scion and rootstock seeds were sown into 98- and 72-cell plug trays filled with a mixed seedling substrate (peat moss and pearlite at a volume ratio of 3:1) on the same day, respectively. After the sowing, the plug trays were covered with plastic film and placed in a germination room at a temperature of 30 °C for two days to promote germination. Then, they were moved to a greenhouse with a temperature of 18–28 °C, a relative humidity of 60–85%, and ordinary light. The plants were fertilized with a water-soluble fertilizer (Product number: 20-10-20 + TE, 1000 times liquid, Shanghai Yongtong chemical Co., Ltd., Shanghai, China).

### 2.2. Device Description and Productive Process

The 2TJGQ-800 grafting machine is composed of a seedling loading platform, clamping and handling mechanism, cutting mechanism, clip-feeding mechanism, clip-sequencing and supply device, conveyor belt, and control system, as shown in Figure 1A. This machine needs two people to operate it and supply the seedlings. It then automatically completes the seedling supplying, holding, cutting, joining, and clipping procedures, as shown in Figure 1B. Its operators are required to pass the grafting machine operation training for more than 48 h, and be proficient in seedling supplying, cutting angle adjusting, clip handling, and blade replacing. The technical requirements for supplying artificial seedlings include adjusting the cotyledon orientation and stem height of the seedlings and keeping the seedlings upright.

The working process was as follows: (1) The seedlings (rootstock and scion) were manually taken out of the plug tray and placed on the seedling loading platform of the grafting machine. (2) The clamping and handling mechanism held the seedlings (rootstock and scion) and transported them to the cutting station. The cutting mechanism cut the seedlings. The rootstock cut off a cotyledon and growth point, and the scion was cut off at 15 mm from the hypocotyl. (3) The clamping and handling mechanism continued to transport the seedlings (rootstock and scion) to the docking station, and the cutting of the rootstock and scion was jointed. (4) The clip-feeding mechanism pushed out a grafting clip to fix the cuts of the rootstock and scion, and then the grafted seedlings were

released and dropped onto the conveyor belt to complete a grafting cycle. The height and cotyledon direction of the seedlings needed to be controlled by a worker during the seedling supplying, and the uniformity of seedlings was selected manually for the grafting machine. In addition, the cutting angle of the rootstock and scion was adjusted to induce the cutting fit effect of the two to reach a perfect state.

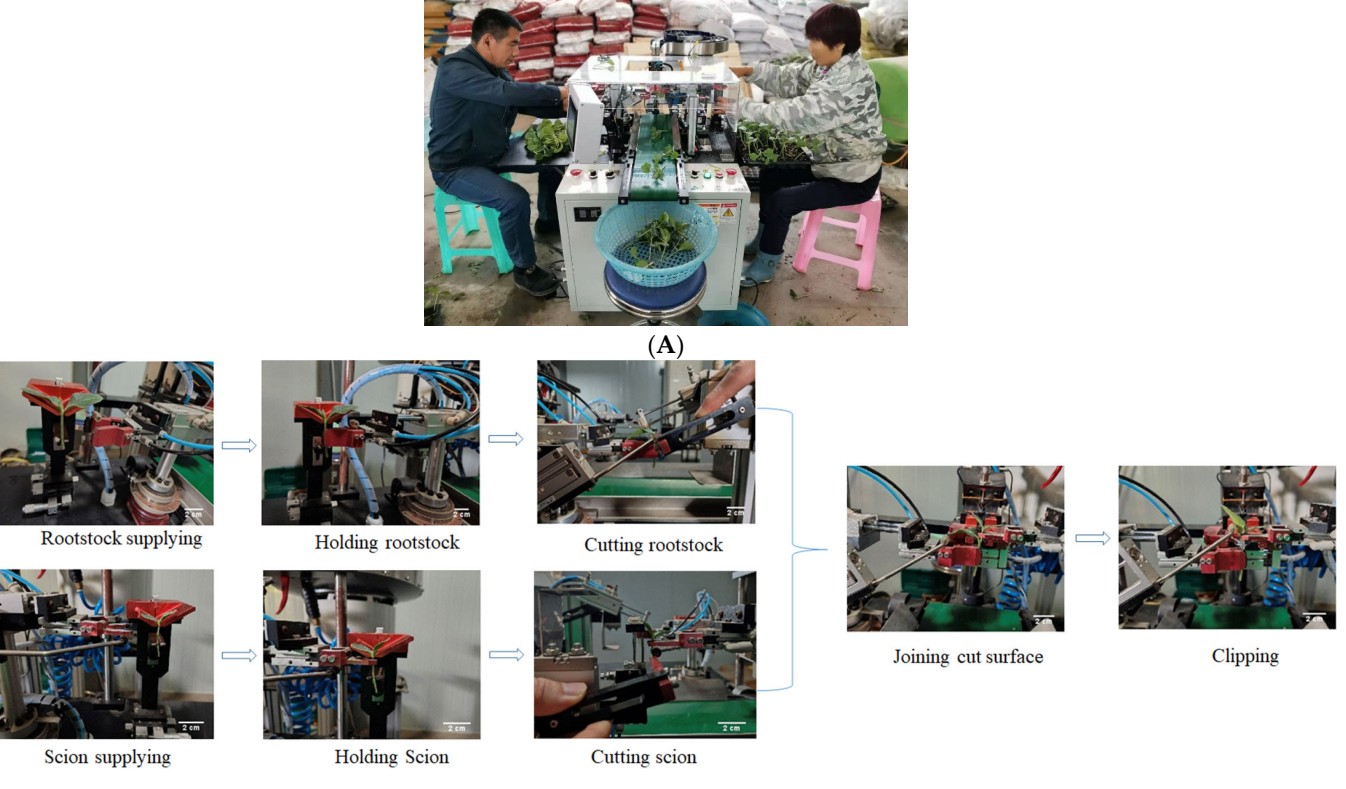

**Figure 1.** (**A**) 2TJGQ-800, launched by the National Agricultural Intelligent Equipment Engineering Technology Research Center in China, a semi-automatic grafting machine used in this study. (**B**) Machine grafting operation process.

### 2.3. Experimental Design

The plants were considered mature and ready for grafting when they had one true leaf, as shown in Figure 2A. The diameters in the area close to the cut varied from 1.58 to 1.75 mm for the scion and 2.71 to 2.80 mm for the rootstock.

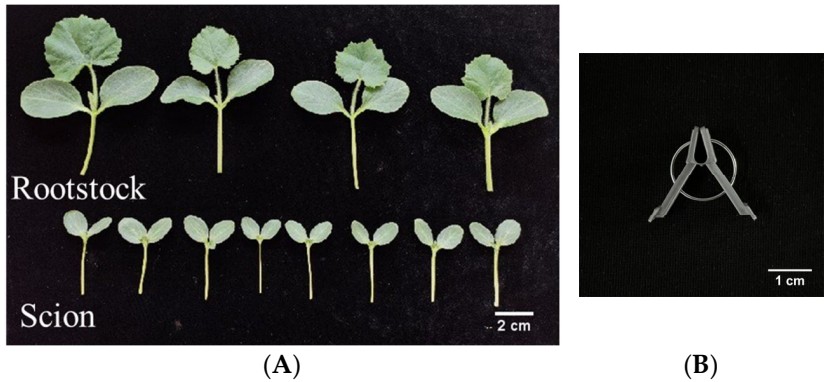

**Figure 2.** (**A**) The morphology of rootstock and scion at grafting, and (**B**) grafting clip for experiments.

The experiments were set up as one-way experiments with three replicates, and each replicate had prepared 500 plants. The root-pruned one-cotyledon grafting was conducted by day 15 after the rootstock and scion seeds had been sown, which was on 24 November 2022 [36]. During the grafting operation, three grafting treatments were established: machine grafting using 2TJGQ-800 (machine), hand grafting by two unskilled grafters (unskilled,) and hand grafting by two skilled grafters (skilled). In addition, the clip was dedicated to the grafting machine. As shown in Figure 2B, it was a one-piece PVC plastic clip. The diameter of the steel wire was 0.7 mm. The preload pressure on the clip mouth was generated by the steel ring. The tail of the clip was squeezed to make the clip open. The procedure of clip was: (1) the tail of the clip was squeezed, (2) the incision of the grafted seedling was put into the clip mouth, and (3) the tail of the clip was released.

The experimental procedure was carried out as follows. First, six workers were chosen and divided into three groups. The first group chosen for the "machine" treatment needed to possess good command of a grafting machine. The operators was required to pass the grafting machine operation training for more than 48 h, and was proficient in seedling supplying, cutting angle adjusting, clip handling and blade replacing. Technical requirements for artificial seedling supplying include adjusting the cotyledon orientation and stem height of seedlings and keeping the seedlings upright. The second group chosen for the "skilled" treatment needed to have more than 2 years grafting experience. The third group chosen for the "unskilled" treatment needed to understand the grafting process and have a small amount of grafting experience (1–2 previous experiences). Secondly, the 500 rootstock plants and scion seedlings were cut off from the stem base. Then, the grafting was completed by the three groups of workers, respectively. Supplying the seedlings to the grafting machine needed two workers, one worker to supply the rootstock seedlings and the other worker to supply the scion seedlings. The hand-grafting work involved cutting, joining, and picking. The two workers cut 300 plants of the scion and rootstock seedlings, respectively. Additionally, they completed the joining and picking alone. Next, the grafting plants were transported into a 72-hole plug tray filled with a mixed seedling substrate.

*2.4. Grafting Seedling Healing and Cultivation*

Immediately after being transplanted, the plants were placed under a plastic film with a day/night cycle of 28 °C/18 °C and more than a 90% humidity in a low-light intensity (75 $\mu mol \cdot m^{-2} \cdot s^{-1}$, 12/12 h photoperiod) environment. The grafted plants were exposed to the air for 1–3 h per day until the scions were alive and had grown normally. After 10 days, the grafted seedlings were transferred to a greenhouse, following common practice.

*2.5. Grafting Efficiency*

The grafts obtained via the "machine" treatment were counted after 30 min of grafting. The time taken to graft 300 plants by hand was also counted. The measurement was conducted 3 times independently.

The grafting speed was investigated using the formula.

$$\text{Grafting speed (plants} \cdot \text{h}^{-1}) = (\text{the number of grafting plants/grafting time}) \tag{1}$$

After the grafting, the number of successfully grafted seedlings was calculated. The rootstock and scion incisions of the successfully grafted seedlings had to be joined together. After turning the grafted rootstocks upside down and shaking them, they were considered to be successfully grafted seedlings if the scion did not fall off. The successfully grafted seedlings are shown in Figure 3.

The grafting success rate was investigated using the formula.

$$\text{Grafting success rate (\%)} = (\text{the number of grafting success seedling/the number of grafting seedling}) \tag{2}$$

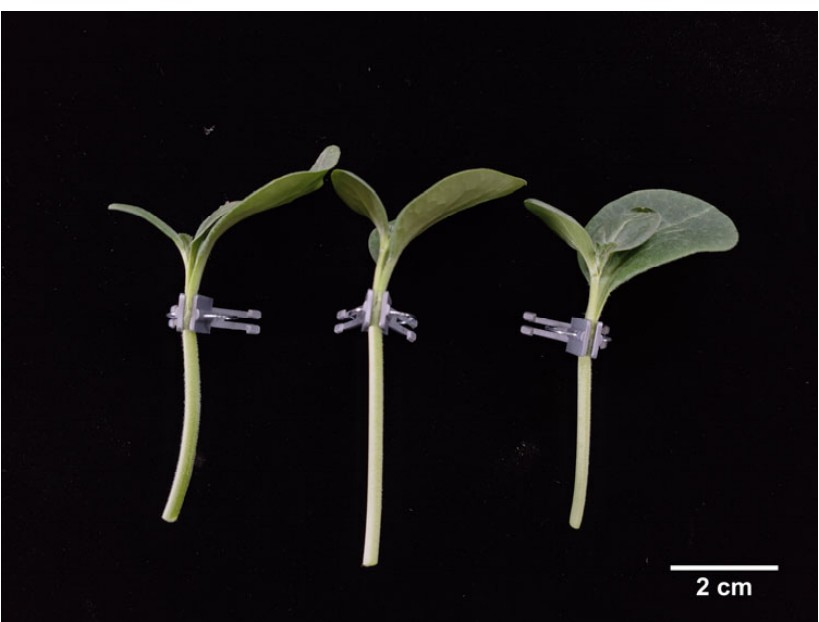

**Figure 3.** The successfully grafted seedlings.

*2.6. Calculation of Hypocotyl Height, Incision Length of Scion and Rootstock*

The hypocotyl height, incision length of the scion, and incision length of the rootstock were measured to analyze the uniformity of the grafting seedlings. After randomly selecting 100 plants of the rootstock and scion seedlings, the height of the hypocotyl and incision lengths of the scion and rootstock were measured using a digital display caliper. The hand-grafting treatment, height of the hypocotyl, and incision lengths of the scion and rootstock were measured after the workers had cut off the rootstock and scion.

When measuring the height of the hypocotyl and incision lengths of the scion and rootstock for the "machine" treatment, the feeding clip system was shut. During the grafting, the workers regularly supplied rootstock and scion seedlings. The cutting mechanism's normal operation and only cutting off the rootstock and scion seedlings without supply the clip, and that were used to measure the incision lengths of the scion and rootstock. This ensured the accuracy of the measurements and avoided a change in the incision lengths of the scion and rootstock as a result of clipping.

The coefficient of variation reflects the uniformity of the parameter. It was investigated using the formula.

$$\text{Coefficient of variation} = \text{standard deviation}/\text{mean}. \tag{3}$$

*2.7. Grafted Survival and Rootstock Regrowth Rate Measurement*

The survival rates of the grafted plants and the rootstock regrowth rates were assessed 10 days after the grafting [17,37]. The grafted plants were considered to have survived if the scion leaves and rootstock stem were turgid, whereas severely wilted scion leaves and stems of both the scion and rootstock were considered as graft failures. Rootstock regrowth (squash leaves) reflected whether the rootstock axillary bud, at the base of the rootstock cotyledon, was removed completely (Figure 4).

$$\text{Survival rate (\%)} = (\text{survival number}/\text{total number of grafted plants}) \times 100\%. \tag{4}$$

$$\text{Rootstock regrowth rate (\%)} = (\text{regrown rootstock number}/\text{total number of grafted plants}) \times 100\%. \tag{5}$$

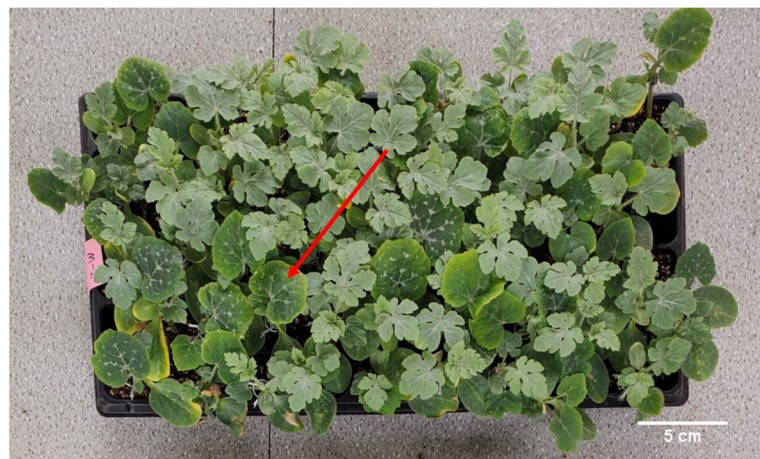

**Figure 4.** The rootstock regrowth from grafting watermelon seedlings. The squash leaves indicated by the red arrow show the rootstock regrowth.

### 2.8. Statistical Analysis

All the data were statistically analyzed by analyses of variance, and the significant differences were determined based on Duncan's tests and were indicated by different letters ($p$ = 0.05) using the SAS 9.0.2 software (SAS Institute Inc., Cary, NC, USA).

## 3. Results

### 3.1. Grafting Efficiency

The "machine", "unskilled", and "skilled" grafting treatments showed significant effects on the grafting speed (Figure 5A). The machine grafting speed was 774 plant·h$^{-1}$. Compared to the "unskilled" and "skilled" grafting treatments, the grafting speeds of the "machine" treatment were 1.65- and 2.55-fold higher. There was a significant difference in the grafting speeds between the hand-grafting treatments ("unskilled" and "skilled" grafting treatments). The grafting speed of the "skilled" grafting treatment increased by 40.53% compared to the "unskilled" grafting treatment. Thus, the grafting speed of skilled workers is more advantageous. From the perspective of grafting speed, machine grafting can compensate for the problem of insufficient labor in production. With the continuous upgrading of automatic technology, it will replace manual seedling supply and the grafting speed will be doubled, so the machine grafting speed can be accepted by users.

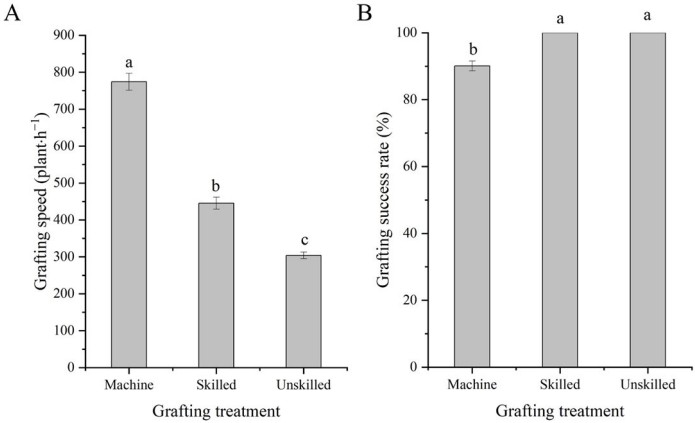

**Figure 5.** Grafting speed (**A**), and grafting success rate (**B**) were measured at grafting. Machine, unskilled, and skilled represent machine grafting with 2TJGQ-800, hand grafting by unskilled grafter, and hand grafting by skilled grafter, respectively. The different small letters indicate significant difference at $p$ = 0.05 levels.

There was no significant difference in the grafting success rates between the hand-grafting treatments (Figure 5B), because one-cotyledon splice grafting is simple and manual grafting can accurately complete the grafting operation. The grafting success rate with hand grafting was 100%, whereas the grafting success rate significantly decreased when using machine grafting. The grafting success rate decreased from 100% to 90.07%. Through improving the standardization of the seedling cultivation and the operation accuracy of the grafting mechanism, the indicators of the machine grafting will be significantly improved.

The reason for the machine grafting failure was statistics. Incomplete rootstock cutting (Figure 6A), grafting seedlings drop failure (Figure 6B), and clip supply failure (Figure 6C) were the three main factors resulting in machine grafting failure. The failure rate caused by the three factors reached 88.89% for all the failures (Figure 6D). Among these, low rootstock positioning resulting in incomplete rootstock cutting accounted for 54.77%. Large rootstock cotyledons resulting in grafting seedlings getting stuck in the clip-feeding mechanism accounted for 16.95%. The clip supply failure accounted for 17.17%, because the attitude of the clip was not in a horizontal state before the clipping operation, resulting in the clip getting stuck in the slide. Other factors resulting in grafting failures accounted for 11.11%, including bent stems of the scion seedlings resulting in the scion stems being too short and the excessive cutting of the rootstock's two cotyledons due to the high position of the rootstock, both of which could not achieve the successful clipping of the grafting seedlings. It can be seen that accurately controlling the position of rootstock seedlings can solve most of the problems causing grafting failure. Achieving a stable and reliable clipping supply by improving and optimizing the clip-feeding mechanism is the second key factor to improving the grafting success rate. In addition, controlling the age of rootstock seedlings and artificially straightening the stem bending of scion seedlings are the keys to improving the accuracy of machine grafting and cutting operations.

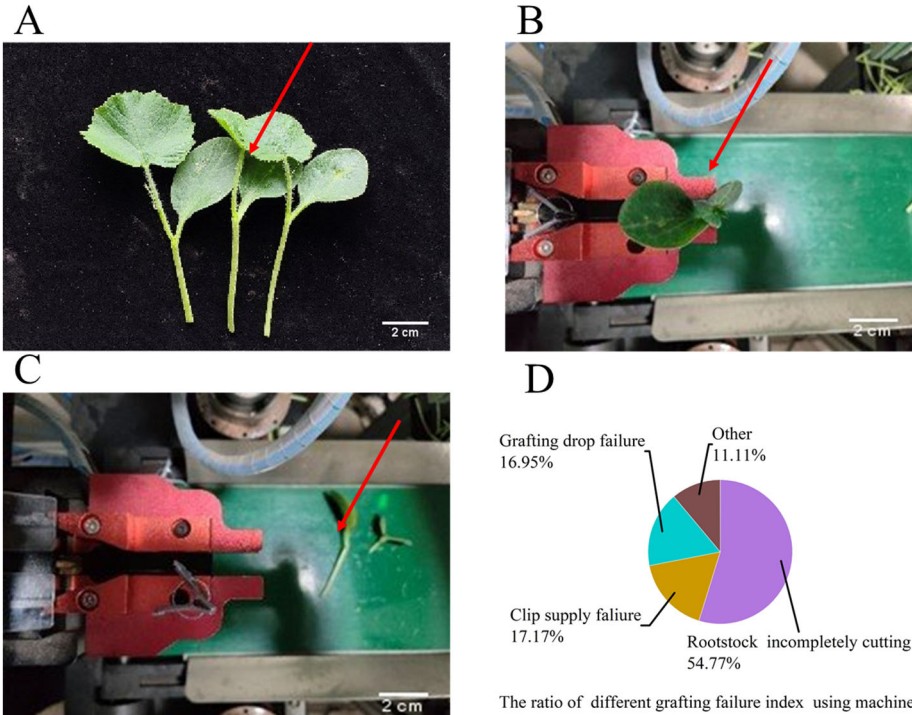

**Figure 6.** The picture of three main factors leading to machine grafting failure (**A**–**C**), and the ratio of different grafting failure indexes using a machine were measured (**D**). Pictures (**A**–**C**), respectively, represent incomplete rootstock cutting, grafting drop failure, and clip supply failure.

### 3.2. Measurement of the Scion Hypocotyl Height, Scion Incision Length, and Rootstock Incision Length at Grafting

The scion hypocotyl heights (Figure 7A), scion incision lengths (Figure 7B), and rootstock incision lengths (Figure 7C) of 100 plants were measured before the clipping. The coefficient of variation reflects the uniformity of the parameter. The smaller the coefficient of variation, the higher the uniformity. This result showed that the coefficients of variation for the scion hypocotyl height, incision length of the scion, and rootstock cutting carried out by the machine were 0.049, 0.089, and 0.068, respectively. The coefficients of variation for the parameters of the cutting carried out by the machine were the smallest. Compared to the "machine" grafting treatment, the coefficients of variation for the scion hypocotyl height treated by the "skilled" and "unskilled" grafting were 1.85- and 4.36-fold higher; those for the incision length of the scion were 1.22- and 1.76-fold higher; and those for the incision length of the rootstock were 1.22- and 1.87-fold higher, respectively.

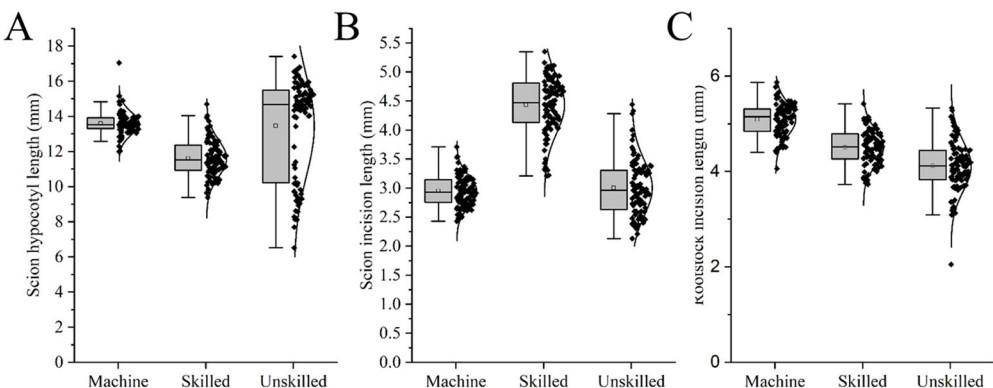

**Figure 7.** The scion hypocotyl height (**A**), scion incision length (**B**), and rootstock incision length (**C**) of 100 plants in each treatment were measured using a digital caliper at grafting. Machine, unskilled, and skilled represent machine grafting with 2TJGQ-800, hand grafting by unskilled grafter, and hand grafting by skilled grafter, respectively.

### 3.3. The Grafted Survival and Rootstock Regrowth Rate

The growth of the grafted watermelon seedlings after healing is shown in Figure 8A. There were no significant differences in the grafting survival rates among the "machine", "unskilled", and "skilled" grafting treatments (Figure 8B). The grafting survival rate of the "machine" treatment was 94.91%. Inappropriate rootstock incision, a mismatch of rootstocks, and scion incision can cause the failure of grafted seedling survival. The grafting survival rates of the unskilled and skilled workers were 97.15% and 97.67%, with a difference of 0.52 percentage points. Manual grafting can accurately complete the grafting work of rootstocks and scions and there is rarely incomplete cutting and joining, so the survival rate of manual grafting is slightly higher than that of machines. However, with the improvement in the cutting accuracy and intelligence of the grafting machine, this grafting survival rate will be improved.

There were no significant differences in the rootstock regrowth rates between the hand-grafting treatments (Figure 8C). The rootstock regrowth rates were 17.95–18.92%. However, the rootstock regrowth rate of the machine grafting reached 72.69%, significantly higher than that of the hand grafting ("unskilled" and "skilled" grafting treatments). An incomplete removal of rootstock growth points is the main reason for the high rate of the machine grafting rootstock regrowth rate. During production, there were morphological differences in the rootstock seedlings. The robot cannot adjust the cutting parameters based on the rootstock seedlings' morphological features and the position of the rootstock seedling supply. Compared to the machine grafting, the regeneration rate of the artificially grafted rootstock was relatively well controlled, because artificial cutting is based on the base of the growth point and its cutting success rate is high.

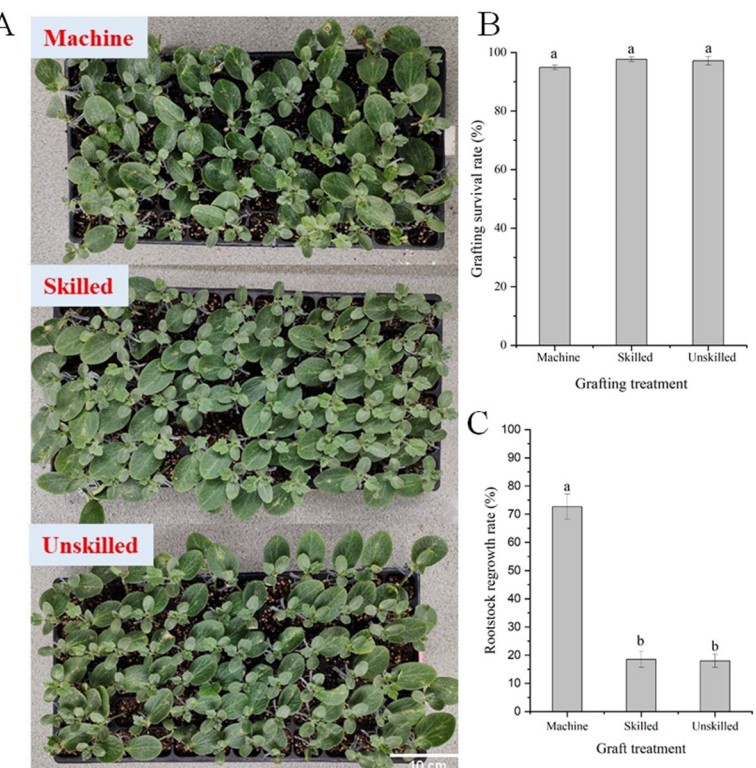

**Figure 8.** Growth of grafted watermelon seedlings after healing (**A**), grafting survival rate (**B**), and rootstock regrowth rate (**C**) were measured at 10 days after grafting. Machine, unskilled, and skilled represent machine grafting with 2TJGQ-800, hand grafting by unskilled grafter, and hand grafting by skilled grafter, respectively. The different small letters in picture (**B**,**C**) indicate significant difference at $p = 0.05$ levels.

It can be seen that the cutting angle consistency of the machine grafting was good and that the accuracy of the artificial seedling supply and cutting parameters were the keys to improving the grafting success rate and grafted seedlings' quality. Hand grafting had advantages in its grafting survival rates and rootstock regrowth rates, but it had no advantage for large scales of production.

## 4. Discussion

Manual grafting is a labor-intensive operation that is highly susceptible to human error. The development of a grafting machine has been considered as an effective alternative to manual grafting. Since the 1980s, researchers have conducted a significant amount of research on vegetable grafting machines [1]. Currently, there are some semi-automatic grafting machines that are relatively mature. However, compared to hand grafting, there were few advantages for grafting efficiency using machine grafting; as such, seedling companies in China cannot accept these machines at present. With the increasing demand for grafted seedlings, the demand for a grafting machine is becoming increasingly urgent. In order to ensure that a grafting machine can be implemented in China as soon as possible, this study was conducted to examine the grafting efficiency and grafting seedling quality of a grafting machine compared to those of skilled grafters and unskilled grafters, based on the grafting machine 2TJGQ-800.

### 4.1. Comparison of the Grafting Speed

One-cotyledon grafting is the simplest and most frequently used method when using a grafting machine [14,15]. The grafting machine in this experiment, 2TJGQ-800, also uses this method. During machine grafting, the grafting process includes feeding seedlings, cutting, joining the cut area of the rootstock and scion together, clipping, and transplantation. A

simple grafting machine can produce 600 grafts per hour with two operators [38]. Due to improvements in grafting machines, the grafting machine in this study can produce 774 grafts per hour. The grafting speed was 1.65–2.55 fold higher than that of the hand grafting. Through training, workers can improve their grafting speed, but the grafting speed remains slower than machine grafting. Consequently, a grafting machine has an advantage over hand grafting in terms of speed.

In addition, manual grafting is labor intensive. During nursery production, grafters must work continuously for 8–10 h every day. With an increase in working time, the grafter will begin to feel tired and inattentive, and the grafting success rate and grafting speed will decrease. However, machine grafting is labor saving. During the machine grafting process, it only needs workers to supply the rootstock or scion seedlings to the machine. Manual grafting requires more work for the cutting, joining, and clipping of seedlings. Thus, the grafting success rate and grafting speed of a machine are stable and scarcely affected by working hours.

### 4.2. Comparison of the Grafted Quality

The grafting machine contains a clamping and handling mechanism, cutting mechanism, clip-feeding mechanism, clip-sequencing and supply device, conveyor belt, and control system. When grafting, the workers place the rootstock and scion seedling in the right position, and the cutting and clip mechanisms are operated based on the operating parameters. These devices ensure that the cutting position and angle are the same. However, as grafters generally cut seedlings based on experience, the cutting positions and angles of the seedlings were random. Thus, the accuracy and uniformity of the machine grafting were higher than those achieved via the hand grafting.

The results showed that the scion hypocotyl height, scion incision length, and rootstock incision length grafted by the machine were 13.5 mm, 2.8 mm, and 5.2 mm, respectively. The coefficients of variation for these parameters were 0.049, 0.089, and 0.068, respectively. Compared to the grafting by the skilled and unskilled grafters, the coefficients of variation for the parameters cut by the machine were the smallest. This highlights the standardized characteristics of a grafting machine. However, the scion incision length cutting completed by the machine was 2.8 mm. This is smaller and not optimized [39]. Thus, it is necessary to further adjust the cutting parameters. The setting of cutting operation parameters is crucial to improving the quality of machine grafting, and it can improve the adaptability of a grafting machine to seedlings.

### 4.3. Comparison of the Grafting Success Rate and the Grafting Survival Rate

In this study, compared to hand grafting, the grafting success rate of the machine grafting decreased from 100% to 90.07%. The grafting success rate was high because the working time of the grafter was under 2 h. The grafters were concentrated and able to correct any mistakes immediately. However, with an increase in working time, the grafter will begin to feel tired and inattentive, and the grafting success rate will decrease.

The machine grafting success rate was lower because the machine operation parameters were set before the grafting and could not be adjusted by changes in the seedling morphological characteristics. The reason for the machine grafting failure was statistics. The failure rate caused by incomplete rootstock cutting was 54.77%. This incomplete rootstock cutting may have been caused by the cutting parameters of the cutting mechanism being inaccurate, an unstable positioning of the seedlings supplied by the worker, or the cutter operation trajectory failing to completely meet the requirements for cutting the rootstock. Though optimizing the machine operating parameters, the machine grafting success rate would increase. Furthermore, the rootstock regrowth rate would decrease.

The grafting survival rate showed no significant differences between the machine grafting and hand grafting. The grafting survival rates for the "machine", "unskilled", and "skilled" grafting treatments were 94.91%, 97.15%, and 97.67%, respectively. The rootstock regrowth rate of the machine grafting reached 72.69%, which was significantly higher than

that of the hand grafting (17.95–18.92%). This result is associated with the accuracy of the cutting device and also conforms to the actual situation. A higher rootstock regrowth rate needs extra labor input to remove it. Consequently, it is necessary to solve the problem of rootstock cutting precision.

### 4.4. Performance Comparison of Similar Grafting Machines

Machines similar to the one described in this study include GR803-U, launched by ISEKI Co., Ltd. in Japan [22] (700 Umaki-cho, Matsuyama-shi, Ehime-ken, 799-2692 Japan), and AFGR-800CS, launched by Helper Robotech Co., Ltd. in Korea (93, Hitech-ro, Jinrye-myeon, Gimhae-si, Gyeongsangnam-do, Korea) [23,24]. Both need two people to supply the seedlings; each job cycle can produce one individual grafting plant. In comparing the performance parameters, the grafting speeds of 2TJGQ-800, GR803-U (Figure 9A), and AFGR-800CS (Figure 9B) were 774 plants·h$^{-1}$, 800 plants·h$^{-1}$, and 800 plants·h$^{-1}$, respectively. The grafting survival rates were 94.07%, 95%, and 95%, respectively. The machine grafting speed depends on the proficiency of the operator. With an improvement in the operator's skill and experience, the grafting speed and efficiency can also be improved. Moreover, the cost of the 2TJGQ-800 grafting machine was CNY 150,000 ¥, amounting to approximately one quarter of that which the same kind of grafting machine costs. Therefore, the 2TJGQ-800 grafting machine has the advantage of market application in terms of both its performance and price.

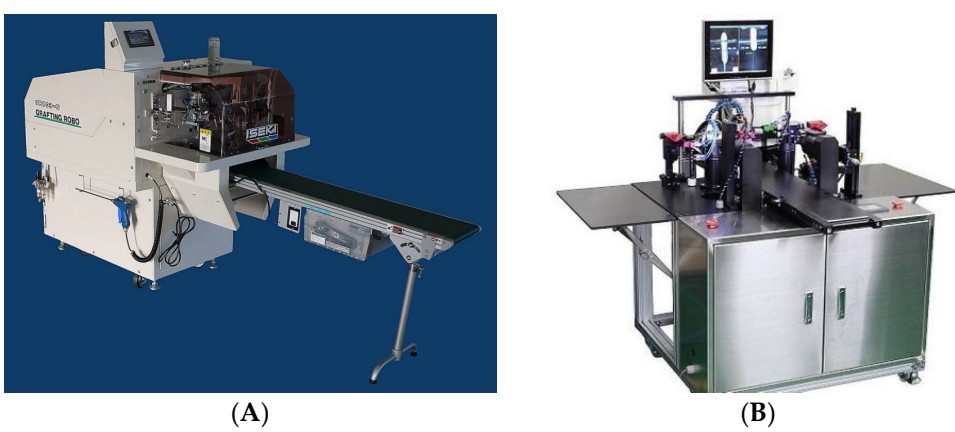

| **(A)** | **(B)** |

**Figure 9.** Similar grafting equipment. (**A**) GR803-U, and (**B**) AFGR-800CS.

### 4.5. Estimation of Operation Cost

The intensive labor inputs of grafting by hand are the main reason for the high cost of grafted seedlings [15]. These labor costs are dependent on grafting speed, success, and the hourly wage of trained employees. The wage used for grafters was CNY 0.08 ¥·plant$^{-1}$ and the wage used for machine operators was CNY 15 ¥·h$^{-1}$. For simplification purposes, a consistent production speed was assumed in the cost simulation. It is estimated that an experienced worker can graft 220 plants when working for 12 h every day. The grafting machine can produce 774 plants·h$^{-1}$ when working for 24 h every day. Thirteen grafters working for 30 days can produce 1 million grafted seedlings. The cost of these grafters is CNY 82,368 ¥. Two machines working for 27 days can produce 1 million grafted seedlings. Six workers in three groups were required to take turns using the machine. The cost of machine operators is CNY 38,880 ¥. Comparatively, the cost of manual grafting is 1.12 times higher than that of machine grafting. In other words, producing 1 million grafted seedlings with a machine can save CNY 43,488 ¥ in labor costs. The cost of a grafting machine is approximately CNY 150,000 ¥. Compared to hand grafting, the labor costs saved in producing 3.45 million grafted seedlings was approximately equal to the cost of purchasing the grafting machine.

## 5. Conclusions

In conclusion, grafting machines have advantages in their grafting speed and uniformity. There were no significant differences in the grafting survival rates between the machine grafting and hand grafting. However, the lower rate of successful grafting and higher rate of rootstock regrowth with the machine grafting, compared to that of the hand grafting, are the main causes for the reduced adoption of grafting machines. In future, more research should be conducted to achieve higher accuracy and efficiency in machine grafting.

Firstly, the accuracy and adaptability of grafting machines will be upgraded. The grafting object is seedlings, which are variable. By improving the adaptability of the machine to the seedlings and the accuracy of the cutting device, the advantages of machine grafting will become apparent. It is necessary to add machine vision using machine vision technology and machine learning algorithms.

Secondly, it is necessary to develop seedling agricultural standards that match the grafting machine. This is beneficial for the standardized operation of machine grafting and realizing the efficient operation of a grafting machine.

Thirdly, a high-efficiency grafting machine is required. Compared to manual grafting using the hole-insertion method, which is mostly used in China, the production efficiency of grafting machines has little advantage. The innovative research and development of plant synchronous intelligent grafting mechanisms and intelligent control technology for the grafting parameters needed to break through the seedling morphological differentiation are the two difficult problems in future research.

Fourth, unmanned or flexible manpower grafted seedling production systems are under consideration. Grafting is a systematic project including a lot of links, such as sowing, grafting, healing, packing, handling, and transporting, etc. When grafting using workers, these seedling links are performed in the seedbed. In other words, grafting with a human needs less extra work. When using grafting robots, extra workers are needed to repeatedly handle the seedlings from the seedbed to the grafting robot. This process is very arduous and requires many workers to participate. Intelligent machinery and equipment will be needed to build an unmanned or flexible manpower grafted seedling production system.

**Author Contributions:** Conceptualization, H.L., K.J. and X.S.; methodology, H.L.; validation, J.Z., M.G. and D.W.; formal analysis, K.L.; investigation, H.L., J.Z., D.W., M.G. and K.L.; data curation, H.L., M.Z. and Y.S.; writing—original draft preparation, H.L. and K.J.; writing—review and editing, Q.Z., K.J. and X.S.; supervision, X.S.; funding acquisition, K.J. All authors have read and agreed to the published version of the manuscript.

**Funding:** This research was supported by the BAAFS Innovation Ability Project (KJCX20220403), the National Nature Science Foundation of China (Grant No. 32171898), the Knowledge Innovation Program of Wuhan-Shuguang Project (2022020801020418), the Key R&D projects in Hubei Province (2021BBA239), the Xiangyang Research and Development Project (2022ABA006997) and the China National Agricultural Research System (CARS-25-07).

**Data Availability Statement:** The data presented in this study are available upon demand from the correspondence author at (jiangk@nercita.org.cn).

**Conflicts of Interest:** The authors declare no conflict of interest.

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
