# Peer review of "A Comparative Analysis of the Grafting Efficiency of Watermelon with a Grafting Machine"

_horticulturae, doi:10.3390/horticulturae9050600_

Round 1
Reviewer 1 Report
Comparative Analysis of the Grafting Efficiency of Watermelon with Grafting Machine by Huan Liang, Juhong Zhu, Mihong Ge, Dehuan Wang, Ke Liu, Mobing Zhou, Yuhong Sun, Qian Zhang, Kai Jiang and Xianfeng Shi presents an analysis of experimental data on the comparison of manual and the machine method of grafting watermelon scion onto a pumpkin rootstock.
The manuscript contains all the sections required for this type of publication. However, the Conclusion section does not contain conclusions and future prospects of the study, but a discussion of the importance of the results obtained, with partial repetitions. This should be changed.
The disadvantage of this manuscript is the lack of scale bars in Figures 2,3,4,6,8. Please fix this by adding scale.
Of the fundamental issues, it seems to me that a description and a characteristic for clips should be added. Also, the authors should discuss other existing machines and robotic techniques in detail. I think that it is advisable to place a link to the patents used and the device scheme.
It seems to me appropriate to expand the description of rootstocks and the features and effectiveness of their use. It is advisable to justify the use of these rootstocks (with references) with the description of their advantages.
The manuscript may be accepted after these corrections have been made.
Author Response
May, 2023
Dear Editors and Reviewers,
On behalf of all the authors, I would like to sincerely appreciate your valuable comments on the manuscript entitled ‘Comparative Analysis of the Grafting Efficiency of Watermelon with Grafting Machine 2TJGQ-800’. Based on your review comments, we have revised the manuscript accordingly and highlighted the changes. In the following, we described the changes we made corresponding to each comment.
Q1. The Conclusion section does not contain conclusions and future prospects of the study, but a discussion of the importance of the results obtained, with partial repetitions. This should be changed.
Authors’ Response: After careful thinking, we revised the conclusion again. Please see line 461.
Q2. The disadvantage of this manuscript is the lack of scale bars in Figures 2,3,4,6,8. Please fix this by adding scale.
Authors’ Response: Thank you for your valuable advice. We add the scale bars in Figures 2,3,4,6,8.
Q3. Of the fundamental issues, it seems to me that a description and a characteristic for clips should be added.
Authors’ Response: A description and a characteristic for clips was added. Please see line 172-175, and Figure 2B.
Q4. The authors should discuss other existing machines and robotic techniques in detail. I think that it is advisable to place a link to the patents used and the device scheme.
Authors’ Response:The one-cotyledon splice grafting method was the only method used by the commercial grafting machines. The operation process in seedlings supplying, seedling clamping, cutting and joining was basically similar. So, the characteristics of grafting machine were not added. In the manuscript, we compare and analyze the performance of two typical grafting machines developed in Japan and South Korea, mainly including the grafting speed, success rate and equipment cost. In addition, we supplement the machine pictures. Please see line 75-84, 87-90, 92-98, and Figure9.
Q5. It seems to me appropriate to expand the description of rootstocks and the features and effectiveness of their use. It is advisable to justify the use of these rootstocks (with references) with the description of their advantages.
Authors’ Response: The rootstock used in this research, was selected based on 2 years comparative analysis on grafting compatible, fruit yield and quality, and disease resistance. The rootstocks have been carried out a large scale of application in Wuhan, China. But, the research related to the rootstock has not been published.

Reviewer 2 Report
1. The present research main question is to compare machine grafting effect on watermelon quality and production compared with traditional methods
2. The topic not original but very important and it is relevant in the field, certainly it would allow farmers and industry to more improve grafting process
3. The specificity of the present work lies in the presence of two dimensions of comparison qualitative and quantitative between the two grafting methods, machine and traditional
4. About methodology, there are no comments, but authors can improve more the discussion part by comparing their results with other literature ones, especially with other machine type
The paper is well presented and very clear.
Author Response
May, 2023
Dear Editors and Reviewers,
On behalf of all the authors, I would like to sincerely appreciate your positive comments on the manuscript entitled ‘Comparative Analysis of the Grafting Efficiency of Watermelon with Grafting Machine 2TJGQ-800’. Based on your review comments, we have revised the manuscript accordingly and highlighted the changes. In the following, we described the changes we made corresponding to each comment.
Comments and Suggestions for Authors
Q1: The present research main question is to compare machine grafting effect on watermelon quality and production compared with traditional methods
Q2: The topic not original but very important and it is relevant in the field, certainly it would allow farmers and industry to more improve grafting process
Q3: The specificity of the present work lies in the presence of two dimensions of comparison qualitative and quantitative between the two grafting methods, machine and traditional
Q4: About methodology, there are no comments, but authors can improve more the discussion part by comparing their results with other literature ones, especially with other machine type.
Authors’ Response: We really appreciate your positive and constructive comments on our manuscript. The manuscript was revised carefully based on the comments. Please see Figure 1B, Figure9, and line 136-140, 171-175, 271-285, 324-336, 435-450.

Reviewer 3 Report
- It is a very interesting study comparing manual labor and mechanical operations for watermelon grafting.
- Lines 70-81: In the literature review, a comparison between the different grafting mechanism already developed (manual, robotic and mechanical) could be included.
- In the introduction a more extensive explanation of the main characteristics of the grafting machine used in the study compared to other grafting machines could be added.
- In the M & M section, the relation of the operator with the grafting machine used in the study could be detailed and remarked.
- Figure 1: A more detailed image of the machine could be included. A sequence of images explaining the operations of the machine could be added.
- Lines 148-160: Were the data obtained from the two workers of each group compared (grafting magnitudes: grafting speed, grafting success rate,…)? Were the actuations of the two workers in each group statistically compared?
- Figure 5: Was the difference in grafting speed between manual and mechanical grafting enough to compensate the lack of labor for the operation. A discussion and estimation about this fact could be included.
- Figure 5b: The inexistence of significant differences between skilled and non-skilled workers could be discussed. In the same line, the inexistence of significant differences in other grafting magnitudes between skilled and non-skilled could be discussed.
- The results shown in Figure 6 should be explained.
- Figure 8b: the results shown could be more extensively discussed. The results of graft survival and rootstock regrowth rate should be extensively discussed. Why was the regrowth rate significantly lower for the manual operation?
- Lines 344-357 (Performance Comparison of Similar Grafting Machines): this part could be discussed in the literature review.
- A estimation of operation costs (with manual or mechanical operation) could be considered and included in the manuscript.
- Conclusions (lines 387-393): It is not clear that this first paragraph is a conclusion of the presented study.
Author Response
May, 2023
Dear Editors and Reviewers,
On behalf of all the authors, I would like to sincerely appreciate your valuable comments on the manuscript entitled ‘Comparative Analysis of the Grafting Efficiency of Watermelon with Grafting Machine 2TJGQ-800’. Your comments not only provide constructive suggestions on improving the quality of the manuscript, but also lead us to in-depth thinking of our approaches. We will benefit from them for our future research. Based on your review comments, we have revised the manuscript accordingly and highlighted the changes. In the following, we described the changes we made corresponding to each comment.
Q1. Lines 70-81: In the literature review, a comparison between the different grafting mechanism already developed (manual, robotic and mechanical) could be included.
Authors’ Response: We have supplemented the performance parameters and characteristics of different grafting machines. Please see line 75-84, 87-90, 92-98.
Q2. In the introduction a more extensive explanation of the main characteristics of the grafting machine used in the study compared to other grafting machines could be added.
Authors’ Response: The main characteristics of the grafting machine used in the study was added. Please see line 75-84, 87-90, 92-98.
Q3. In the M & M section, the relation of the operator with the grafting machine used in the study could be detailed and remarked.
Authors’ Response: In the M & M section, the relation of the operator with the grafting machine used in the study was detailed and remarked. Please see line 136-140.
Q4. Figure 1: A more detailed image of the machine could be included. A sequence of images explaining the operations of the machine could be added.
Authors’ Response:The pictures of seedling supply, cutting, joining and clipping for grafting machine operation was added. Please see Figure 1B.
Q5. Lines 148-160: Were the data obtained from the two workers of each group compared (grafting magnitudes: grafting speed, grafting success rate,…)? Were the actuations of the two workers in each group statistically compared?
Authors’ Response: The two workers in each hand-grafting group was not grafting alone. First, cutting the rootstock and scion is carried out separately. Second, the two workers carried out grafting operations separately. The grafting speed and success rate were counted based on the work by 2 workers. Therefore, it is not possible to count and compare everyone's actions. In addition, the work of the two operators of the grafting machine is to supply seedlings, which is synchronized. The machine automatically completes the cutting, joining and clipping operations. So, the grafting speed and success rate of the grafting were also counted based on the work by 2 operators.
Q6. Figure 5: Was the difference in grafting speed between manual and mechanical grafting enough to compensate the lack of labor for the operation. A discussion and estimation about this fact could be included. Figure 5b: The inexistence of significant differences between skilled and non-skilled workers could be discussed. In the same line, the inexistence of significant differences in other grafting magnitudes between skilled and non-skilled could be discussed.
Authors’ Response: Based on your comments, we have supplemented the relevant discussions. Please see line 251-255, 376-395.
Q7. The results shown in Figure 6 should be explained.
Authors’ Response: Based on your comments, the results shown in Figure 6 was explained. Please see line 271-285.
Q8. Figure 8b: the results shown could be more extensively discussed. The results of graft survival and rootstock regrowth rate should be extensively discussed. Why was the regrowth rate significantly lower for the manual operation?
Authors’ Response: Based on your comments, the results shown in Figure 8B and Figure 8C were added. Please see line 312-319, 324-336.
Q9. Lines 344-357 (Performance Comparison of Similar Grafting Machines): this part could be discussed in the literature review.
Authors’ Response: In the literature review, the performance and characteristic of similar grafting machine were added. Please see line 75-84, 87-90, 92-98.
Q10. A estimation of operation costs (with manual or mechanical operation) could be considered and included in the manuscript.
Authors’ Response: In the discussion, the estimation of operation costs with manual and mechanical operation was added. Please see line 435-450.
Q11. Conclusions (lines 387-393): It is not clear that this first paragraph is a conclusion of the presented study.
Authors’ Response: Based on your comments, we have rearranged the conclusion section. Please see line 461-519.

Reviewer 4 Report
The paper ‘Comparative Analysis of the Grafting Efficiency of Watermelon with Grafting Machine’ studies the reliability and feasibility of machine grafting through a comprehensive evaluation of production capacity and grafting seedling quality.
The theme of the manuscript and the obtained results are interesting, anyway some points should be clarified as described in the following lines:
- The device description (pag. 3 Line 113) should be completed with schemes or drawings clearly describing the mechanisms and their mechanical connections;
- Moreover, for the same paragraph, it should be interesting to have more information about the work of the two operating people (time of intervention, risks, abilities requested and so on…);
- About Eq. (3). (pag. 5 Line 195), could you please specify on what data you calculated standard deviation and mean?
- Could you please improve the quality of Figure 6 (D) that is difficulty readable?
- The small letters significant shown in Figure 8 should be clarified in the manuscript or, otherwise, the small letters could be removed.
- In the comparison shown in paragraph 4.4 (line 344) the real advantage of the 2TJGQ-800 machine compared with the other concurrencies seems to be only the cost; please clarify it.
Author Response
May, 2023
Dear Editors and Reviewers,
On behalf of all the authors, I would like to sincerely appreciate your valuable comments on the manuscript entitled ‘Comparative Analysis of the Grafting Efficiency of Watermelon with Grafting Machine 2TJGQ-800’. Your comments not only provide constructive suggestions on improving the quality of the manuscript, but also lead us to in-depth thinking of our approaches. We will benefit from them for our future research. Based on your review comments, we have revised the manuscript accordingly and highlighted the changes. In the following, we described the changes we made corresponding to each comment.
Q1. The device description (pag. 3 Line 113) should be completed with schemes or drawings clearly describing the mechanisms and their mechanical connections; Moreover, for the same paragraph, it should be interesting to have more information about the work of the two operating people (time of intervention, risks, abilities requested and so on…);
Authors’ Response: The pictures of machine grafting operation process was added. The operators was required to pass the grafting machine operation training for more than 48 hours, and was proficient in seedling supplying, cutting angle adjusting, clip handling and blade replacing. Technical requirements for artificial seedling supplying include adjusting the cotyledon orientation and stem height of seedlings and keeping the seedlings upright. Please see line 136-140, and Figure 1B.
Q2. About Eq. (3). (pag. 5 Line 195), could you please specify on what data you calculated standard deviation and mean?
Authors’ Response: Based on your comments, we have supplemented the attached table with the raw data for calculating standard deviation and mean.
Q3. Could you please improve the quality of Figure 6 (D) that is difficulty readable?
Authors’ Response: Thanks for your valuable suggestion, we have readjusted the Figure 6(D).
Q4. The small letters significant shown in Figure 8 should be clarified in the manuscript or, otherwise, the small letters could be removed.
Authors’ Response: Below the Figure 8, we explain the meaning of small letters. Based on your comments, the explanation may not be clear, so we have also revised the explanation. Please see line 343-347.
Q5. In the comparison shown in paragraph 4.4 (line 344) the real advantage of the 2TJGQ-800 machine compared with the other concurrencies seems to be only the cost; please clarify it.
Authors’ Response: Since we cannot own these two grafting machines (GR803-U, AFGR-800CS), we can only use the official data of these two grafting machines for comparison. Compared with the grafting machine used in the research, there is no significant difference on grafting speed and grafting success rate. It shows that the testing machine has reached the technical level of similar equipment in the world. In addition, it has great advantages in cost, which is benefit for applying in China. However, the purpose of this study is to explore whether the grafting machine can replace manual grafting, and the results confirm that the grafting machine has the advantage of replacing manual grafting.

Round 2
Reviewer 1 Report
Article Comparative Analysis of the Grafting Efficiency of Watermelon with Grafting Machine by authors Huan Liang, Juhong Zhu, Mihong Ge, Dehuan Wang, Ke Liu, Mobing Zhou, Yuhong Sun, Qian Zhang, Kai Jiang, Xianfeng Shi are sufficiently corrected, changes made .
The manuscript may be published.
Reviewer 3 Report
The authors have considered and answered all the reviewer´s comments. Some of the improvements of the manuscript need to be implemented with further research. However, the results presented in the manuscript could represent an important contribution in the field.